# Sex Bias in Frailty Screening: A Cross-Sectional Analysis of PRISMA-7 and the Clinical Frailty Scale in Primary Care

**DOI:** 10.3390/diagnostics15070915

**Published:** 2025-04-02

**Authors:** Christian J. Wiedermann, Verena Barbieri, Dietmar Ausserhofer, Adolf Engl, Giuliano Piccoliori, Angelika Mahlknecht

**Affiliations:** Institute of General Practice and Public Health, Claudiana College of Health Professions, 39100 Bolzano, Italyangelika.mahlknecht@am-mg.claudiana.bz.it (A.M.)

**Keywords:** frailty screening, PRISMA-6, PRISMA-7, Clinical Frailty Scale, sex bias, primary care, older adults, frailty prevalence, frailty tools validation

## Abstract

**Background/Objectives**: Frailty screening is essential in primary care for the early identification of vulnerable older adults. PRISMA-7 is a widely used screening tool, but Item 2 (“Are you male?”) introduces potential sex bias and overestimates frailty in men. PRISMA-6, a modified version that excludes Item 2, might provide a more equitable alternative. This study evaluates PRISMA-6’s alignment with the Clinical Frailty Scale (CFS) and its impact on sex-specific frailty classification. **Methods:** A cross-sectional study was conducted in 142 general practices across South Tyrol, including 9190 general practice patients aged ≥75 years. Frailty was assessed using PRISMA-7, PRISMA-6, and the CFS. Correlations between tools were calculated using Kendall’s Tau-b, whereas Fisher’s z-test was used to compare differences in alignment. The frailty prevalence and odds ratios were stratified according to sex and age. **Results:** PRISMA-6 showed a stronger correlation with the CFS (τ = 0.492) than PRISMA-7 (τ = 0.308, *z* = −10.2, *p* < 0.001). This effect was pronounced in men (*z* = −9.8, *p* < 0.001), whereas no difference was observed in women (*z* = 0.00, *p* = 1.000). PRISMA-6 reduced the frailty detection rate in men and was more closely aligned with the CFS. **Conclusions:** PRISMA-6 demonstrated improved alignment with the CFS and reduced sex bias compared to PRISMA-7. However, its use as a screening tool for men requires prospective validation in diverse settings. PRISMA-6 shows promise as a reliable and equitable frailty screening tool and should be considered for use in future studies, particularly in primary care settings, while awaiting further prospective validation.

## 1. Introduction

Frailty is a multidimensional syndrome characterised by a decline in physiological reserve and increased vulnerability to stressors, leading to an elevated risk of adverse outcomes, such as hospitalisation, institutionalisation, and mortality [1]. Given the global aging population, the early identification of frailty has become a priority in healthcare to facilitate timely interventions that maintain autonomy, enhance quality of life, and reduce healthcare expenditure [2,3].

In primary care settings, general practitioners (GPs) are uniquely positioned to identify frailty as part of routine care [2]. Systematic frailty screening is increasingly being recognised as a crucial component of preventive and proactive healthcare [4,5,6]. In Italy, Ministerial Decree No. 77/2022 underscores the necessity of structured frailty screening to identify at-risk older adults and integrate them into tailored care pathways [7]. To meet this objective, simple, validated, and easily administrable tools are essential [6,8].

In South Tyrol, frailty screening in primary care has been historically limited [9,10,11], despite the region’s aging population. Recent initiatives supported by governmental funding and contractual negotiations with GP unions have introduced systematic frailty screening as an integral component of routine care [12]. This endeavour, scientifically guided by the Institute of General Practice and Public Health Bolzano-Bozen, presents an opportunity to evaluate and refine the implementation of instruments such as the Program of Research to Integrate the Services for the Maintenance of Autonomy-7 (PRISMA-7) [13] and the Clinical Frailty Scale (CFS) [8]. Both the CFS and PRISMA-7 are efficacious tools for frailty screening, demonstrating comparable accuracy in diverse settings [14,15,16]. While neither is considered a gold standard, they provide valuable insights into frailty status, facilitating the identification of individuals at risk of adverse outcomes. The selection between these instruments should be predicated on a specific context and population requirement [17].

Among the various frailty assessment tools, PRISMA-7 and the Clinical Frailty Scale (CFS) were selected based on their complementary strengths and feasibility in routine primary care. PRISMA-7 is a self-administered, highly sensitive questionnaire that has been validated in community and primary care populations [13,18,19] and is recommended for its ease of use and quick administration [6]. In contrast, the CFS offers a clinician-judged assessment based on observable physical and cognitive function, providing a structured yet intuitive measure of functional status across nine levels [14,20]. Unlike more complex instruments such as the Fried Frailty Phenotype or the Edmonton Frail Scale, both PRISMA-7 and CFS are practical for implementation in real-world primary care settings without the need for special equipment or extensive training [15,16]. While PRISMA-7 has been criticized for potential sex bias, particularly due to Item 2 (“Are you male?”) [11], its modified version, PRISMA-6, aims to address this limitation while retaining the tool’s overall simplicity. These tools were selected not only for their widespread validation and practicality but also because their differing perspectives—self-report vs. clinician judgment—offer a valuable comparison for evaluating potential sex bias and refining frailty screening practices in older adults.

PRISMA-7, a seven-item questionnaire with a threshold of ≥3 indicating frailty [13], has demonstrated high sensitivity in diverse populations and is widely used in community and primary care settings [18,19,21]. However, various limitations have been identified [22,23], including a potential sex bias which has been noted due to Item 2 (“Are you male?”), which automatically assigns one point to male participants [11]. While this may be intended to reflect higher short-term mortality risk in frail men, it can also lead to disproportionate frailty classification among men and the under-recognition in women, who experience greater loss of healthy life expectancy. This study, therefore, investigates the effect of removing Item 2 by comparing PRISMA-6 with PRISMA-7 to evaluate whether this modification improves the equity and clinical alignment of frailty screening outcomes.

The CFS provides a clinician-administered assessment of frailty across nine levels, encompassing functional and cognitive dimensions [20]. A comparative analysis of self-reported PRISMA-7 [13] and the clinician-assessed CFS presents a valuable opportunity to investigate the congruence between subjective and objective measures of frailty, particularly in identifying sex-based disparities and enhancing screening efficacy.

This study had two objectives.

To evaluate PRISMA-7 with and without Item 2 (“PRISMA-6”) against the CFS in a primary care setting.To assess the impact of Item 2 on sex-specific frailty prevalence.

By examining these aspects, this study aimed to contribute to the optimisation of frailty screening tools, aligning with national healthcare reforms and enhancing elderly care in primary care settings.

## 2. Methods

### 2.1. Study Design

This study was conducted as part of a government-endorsed healthcare project in South Tyrol aimed at systematically screening for frailty among community-dwelling older adults aged 75 years and above. The initiative was implemented within routine general practice and facilitated through contractual agreements between the provincial healthcare system and GPs [12]. The study methodology comprised an initial cross-sectional phase, with the intention of conducting a subsequent longitudinal investigation to examine the progression of frailty and its associated outcomes over a 12-month period. The principal findings of this study will be published separately. This article presents the results of sex-specific frailty prevalence obtained in the cross-sectional phase of the research.

### 2.2. Setting and Participants

This study was conducted across general practices in South Tyrol, where approximately 300 practices provide primary care. All the GPs were invited to participate voluntarily and were offered remuneration for their involvement. This study included community-dwelling individuals who were 75 years of age or above, encompassing those receiving end-of-life care at home. Patients domiciled in institutional settings such as long-term care facilities were excluded. The participating general practitioners were estimated to screen a median of approximately 185 patients aged 75 years or older each, corresponding to the typical demographic profile of older adults in the region.

### 2.3. Screening Instruments, Data Collection, and Management

The screening protocol employed two sequential tools to assess frailty (Appendix A). Initially, the self-administered PRISMA-7 questionnaire was used to identify potential frailty [13]. Patients with a PRISMA-7 score ≥ 3 were categorised as frailty-positive and selected for further clinical evaluation. These patients were subsequently assessed using the CFS, a clinician-administered tool that requires direct in-person evaluation [20]. A CFS score of ≥5 was considered indicative of frailty. To support the consistent application of the CFS, all participating GPs were provided with written guidance and interpretation aids. A help desk at the coordinating centre was available for consultation throughout the screening phase. As the CFS is already used in many clinical settings across Italy, participating GPs were expected to have at least a basic familiarity with its structure and scoring.

The PRISMA-7 questionnaire was distributed to eligible participants for self-completion and was returned using standardized forms. Patients meeting the frailty threshold (score ≥ 3) were subsequently evaluated with the CFS during clinical visits.

In addition to all patients who screened positive on PRISMA-7 (score ≥ 3), some general practitioners also documented a CFS score for patients with PRISMA-7 scores < 3, although this was not explicitly requested by the study protocol. These additional assessments were included in the analysis to explore the diagnostic performance of PRISMA-7 and its potential for misclassification, particularly among patients not flagged as frail by the screening tool.

All the collected data were anonymised and stored securely in databases managed by the Institute of General Practice and Public Health in Bolzano.

### 2.4. Outcome Measures

The primary outcomes of interest were as follows:The strength and alignment of the correlations between PRISMA-7, PRISMA-6, and CFS scores in identifying frailty.The impact of PRISMA-7’s Item 2 (“Are you male?”) on sex-specific frailty prevalence and the diagnostic performance of the tool in comparison with PRISMA-6 and CFS.

### 2.5. Statistical Analysis

No a priori sample size calculation was performed, as this was a population-based study embedded in routine practice with the aim of including all eligible patients aged 75 years and older in the participating general practices. The final sample size (*n* = 9190) reflects approximately one-third of the total population aged 75+ in the region, ensuring high representativeness and statistical power for subgroup analyses.

A PRISMA-6 score of ≥3 was used to classify individuals as frail, consistent with the established threshold applied in PRISMA-7 validation studies and supported by prior research [11]. Descriptive statistics were calculated to summarise the participant characteristics, PRISMA-7 scores, and CFS results. Statistical analyses were conducted using the Jeffreys’ Amazing Statistics Program (JASP; University of Amsterdam, Amsterdam, The Netherlands). Proportions, including prevalence values, were calculated in JASP using contingency tables.

A normality assessment of the score distributions was performed using skewness and kurtosis. Given the large sample size, the Shapiro–Wilk test was not applicable. Since the data were not normally distributed, nonparametric correlation methods were considered. While Spearman’s rank correlation is often used for ordinal data, Kendall’s Tau-B was chosen due to the presence of tied ranks, which are common in frailty scoring.

Confidence intervals (CIs) for proportions were not directly available in JASP and were supplemented with calculations performed in Excel (Microsoft Corporation, Redmond, Washington, DC, USA). The 95% CI for prevalence was calculated based on the standard error of the proportion, using the normal approximation to the binomial distribution. The lower and upper bounds of the CI were derived by subtracting and adding the product of the standard error and critical value for a 95% confidence level.

Odds ratios (ORs) were derived from frailty detection rate values calculated using JASP. The 95% confidence intervals (CIs) were determined in Excel by calculating the standard error of log-transformed odds. The confidence bounds for the log odds were exponentiated to obtain the CI for the odds ratio on the original scale, reflecting the inherent asymmetry of the OR estimates. While prevalence values directly reflect frailty detection rates, ORs were used to provide a relative measure of the likelihood of being classified as frail versus non-frail. The reference group for each OR was the total study population (*n* = 9190), and each tool’s classification was compared to those not classified as frail by the same tool. As frailty prevalence was particularly high with PRISMA-7, ORs may overstate relative differences in such contexts. Therefore, absolute prevalence values are reported alongside ORs to aid interpretation.

Correlations between PRISMA-7, PRISMA-6, and the CFS were calculated using Kendall’s tau-b test. Analyses were conducted for the overall cohort and were stratified according to sex. The strength of the correlations was interpreted based on the standard thresholds, wherein values of 0.1–0.3 indicate weak correlations, 0.3–0.5 moderate correlations, and >0.5 strong correlations, respectively. The agreement between binary frailty classifications (frail vs. non-frail) from PRISMA-7, PRISMA-6, and the CFS was evaluated using Cohen’s kappa coefficient. Fisher’s z-test was performed to compare the strength of the correlations between PRISMA-7, PRISMA-6, and the CFS [24]. Kendall’s tau-b coefficients were transformed into Fisher’s z-values, and the z-statistic for comparing the correlations was calculated as the difference between the two transformed correlations divided by the standard error of the difference. Statistical significance was evaluated using a one-tailed test for positive correlations, with *p* < 0.05 considered significant.

## 3. Results

### 3.1. Study Population Characteristics

A total of 295 GPs in South Tyrol were invited to a government-supported frailty-screening project. Of these, 142 (48.1%) agreed to participate. The median age of the GPs was 47 years (IQR 39–58), with practices almost equally distributed between urban (52.8%) and rural (47.2%) areas. Each GP cared for about 1,600 patients, of whom approximately 190 were aged 75 years and older, consistent with the median number screened. Participant enrolment is shown in Figure 1.

From these practices, 19,501 patients aged 75 years or older participated in the screening, representing 34.1% of the regional elderly population, estimated at 57,200 in 2023. Most participants (57.5%) were female with a median age of 81 years (IQR 78–85 years).

Screening was performed using the PRISMA-7, with 18,658 patients (95.7%) completing the questionnaire (median of 122 patients per GP). Among them, 42.7% (*n* = 7970) were identified as frail (PRISMA-7 score ≥ 3). This subset was further assessed using the CFS. Since additional 1220 patients with PRISMA-7 scores < 3 also underwent CFS scoring, the final study cohort was 9190 patients (mean age: 84.2 years; SD: 5.48; range 75–104) with 4212 men (mean age: 83.6 years; SD: 5.24; range 75–103) and 4978 women (mean age: 84.7 years; SD: 5.63; range 75–104).

### 3.2. Frailty Detection Rate by PRISMA-7, PRISMA-6, and CFS

To assess the distribution of PRISMA-7, PRISMA-6, and CFS scores, their frequency distributions across age groups (<75, 75–84, and ≥85 years) and sex (male, female) were assessed (Figure 1).

The skewness and kurtosis values for PRISMA-7 (skewness = 0.001, kurtosis = −0.49), PRISMA-6 (skewness = 0.010, kurtosis = −0.90), and the CFS (skewness = 0.213, kurtosis = −0.53) suggest that these distributions are approximately symmetrical but slightly platykurtic, indicating a flatter distribution compared to a normal curve.

Figure 2 and Table 1 summarise the frailty detection rates and odds ratios (ORs) across PRISMA-7, PRISMA-6, and the CFS stratified by the sex and age group. PRISMA-7 consistently showed a higher frailty prevalence in males compared to females across all age groups. In the overall study population (including all age groups), PRISMA-7 classified 89.7% of male participants and 84.2% of female participants as frail, with corresponding odds ratios (ORs) of 8.68 and 5.34, respectively. In contrast, the CFS identified 35.3% of male participants and 48.4% of female participants as frail (OR: 0.54 and 0.94, respectively).

PRISMA-6, which eliminates Item 2 of PRISMA-7, showed a lower frailty prevalence in males compared to PRISMA-7. In the overall study population (across all age groups), PRISMA-6 classified 53.0% of male participants (OR: 1.13) and 84.2% of female participants (OR: 5.34) as frail. In the 75–84 years age group, PRISMA-6 identified 40.9% of males (OR: 0.69) and 72.4% of females (OR: 2.62) as frail, while in the 85+ age group, frailty prevalence was 68.3% in males (OR: 2.15) and 95.8% in females (OR: 22.56).

The CFS consistently demonstrated a higher frailty prevalence and OR in females compared to males across all age groups. The current study substantiates that PRISMA-6 not only mitigates sex-related classification bias observed in PRISMA-7 but also shows more consistent alignment with the CFS, despite some variation in prevalence patterns across tools (Figure 3).

### 3.3. Correlations Between PRISMA-7, PRISMA-6, and CFS Classifications

Correlations between PRISMA-7, PRISMA-6, and the CFS were assessed using Kendall’s tau-b correlation (Table 2). PRISMA-7 and PRISMA-6 showed a strong positive correlation. Both PRISMA-7 and PRISMA-6 moderately correlated with the CFS.

The correlation between PRISMA-6 and the CFS (τ = 0.492) was stronger than that between PRISMA-7 and the CFS (τ = 0.308). To further assess agreement in frailty classification, Cohen’s kappa was calculated. In the overall sample, agreement between PRISMA-7 and the CFS was fair (κ = 0.186), identical to the value observed for PRISMA-6 (κ = 0.186). Stratified by sex, kappa values were κ = 0.106 in men and κ = 0.277 in women, again identical for both instruments due to their classification equivalence in women. Fisher’s z-test was performed to assess the statistical significance of this difference. The z-value for the test was −10.2, with *p* < 0.001, indicating a significant difference in correlation strength. These results confirm that PRISMA-6 aligns more closely with the CFS than PRISMA-7.

Stratified analyses revealed that this pattern is particularly pronounced in males. Among men, the correlation between PRISMA-6 and the CFS (τ = 0.576) was notably stronger than that between PRISMA-7 and the CFS (τ = 0.226), with Fisher’s z-test yielding z = −9.8, *p* < 0.001. This suggests that PRISMA-6 provides a more accurate frailty classification for men by mitigating the overestimation observed with PRISMA-7.

In contrast, no difference in correlation strength was observed in females, where both PRISMA-7 and PRISMA-6 exhibited identical correlations with the CFS (τ = 0.389). Fisher’s z-test confirmed a lack of statistical difference (z = 0.00, *p* = 1.000). These findings indicate that the improved alignment of PRISMA-6 with the CFS is driven by its effect on the male subgroup.

## 4. Discussion

Frailty screening is a critical component of proactive healthcare for the elderly, particularly in primary care settings, where early identification can guide timely interventions [5]. This study evaluated the performance of PRISMA-7, a brief self-administered screening tool with high sensitivity, and the CFS, a clinician-administered tool that incorporates functional and cognitive dimensions, in a government-supported frailty screening initiative among older adults in South Tyrol. The findings revealed that PRISMA-7 identified a substantially higher prevalence of frailty than the CFS, consistent with its intended role as a sensitive screening tool. While PRISMA-7 classified 86.7% of the overall population as frail, the CFS identified only 42.4% as frail. The high prevalence rates reported herein are attributable to the selected study population predominantly comprised patients who were frailty-positive for PRISMA-7. Frailty prevalence increased with age, with individuals aged 85 years or older exhibiting a markedly higher frailty prevalence than those aged 75–84 years.

The high frailty prevalence reported in this study, 86.7% using PRISMA-7 and 42.4% using the Clinical Frailty Scale (CFS), reflects the structured two-step screening approach rather than general population rates. In a representative population-based survey conducted in South Tyrol, frailty prevalence among community-dwelling individuals aged ≥75 years was substantially lower: 33.9% with PRISMA-7 and 27.0% with PRISMA-6 [11]. These findings confirm that the elevated detection rates in our study result from the selective application of the CFS to patients who screened positive on PRISMA-7 in alignment with the design of routine screening workflows in primary care. Therefore, Table 1 reports detection rates within the screened population, not population prevalence. This distinction is crucial for interpreting the diagnostic yield of screening tools versus their application in epidemiologic surveillance.

Sex-specific analyses revealed significant differences in frailty prevalence. PRISMA-7 classified a higher proportion of males (89.7%) as frail than females (84.2%), whereas the CFS demonstrated the opposite trend, with more females (48.4%) identified as frail than males (35.3%). This discrepancy reflects PRISMA-7’s Item 2 (“Are you male?”), which assigns an additional point to males and may contribute to the overestimation of frailty prevalence in men. PRISMA-6, a modified version of PRISMA-7 that excludes Item 2 [11], demonstrated reduced frailty prevalence in males (53.0%) while aligning more closely with the prevalence observed using the CFS. These findings correspond with the existing literature that indicates that females generally exhibit higher frailty prevalence due to longer life expectancy, multimorbidity, and greater disability rates [25,26].

Correlation analysis confirmed a strong alignment between PRISMA-7 and PRISMA-6, notwithstanding the removal of Item 2 in the latter. Both instruments exhibited positive associations with the CFS; however, PRISMA-6 demonstrated a significantly stronger correlation. Fisher’s z-test confirmed this difference to be statistically significant, substantiating PRISMA-6’s enhanced alignment with clinician-assessed frailty.

Stratified analyses revealed that this effect was predominantly driven by the male subgroup, wherein PRISMA-6 exhibited a substantially stronger correlation with the CFS than PRISMA-7, supporting the hypothesis that PRISMA-7 Item 2 overestimates frailty in men. These observations underscore PRISMA-6 as a more equitable frailty screening instrument, mitigating sex bias while maintaining strong alignment with clinician assessment. Further research is warranted to validate its predictive value for adverse outcomes and to assess its applicability across diverse populations.

The study by Ausserhofer et al. [11] corroborates these findings, indicating that in community-dwelling elderly individuals, PRISMA-7 overestimated frailty prevalence in males, with 34.7% of men classified as frail compared to 33.0% of women. The omission of Item 2 in the PRISMA-6 reversed this trend, revealing a lower frailty prevalence in men (21.4%) and a stable prevalence in women (33.0%). This reversal highlights the role of Item 2 in potentially introducing a sex bias and inflating frailty prevalence in males. Excluding Item 2 improved the internal consistency of PRISMA-6 (Cronbach’s alpha 0.75) compared to that of PRISMA-7 (0.64), supporting the reliability of the modified tool [11]. The current study substantiates their conclusions, demonstrating that PRISMA-6 not only mitigates this bias but also aligns frailty classification more closely with the CFS, a clinician-administered tool.

General practitioners value tools that are simple to administer and interpret while providing actionable insights [27]. Despite PRISMA-7’s widespread utilisation, there exists scepticism among certain general practitioners regarding its suitability as a standalone screening instrument, particularly in capturing the complexity of frailty in diverse populations [21,28]. Consequently, emerging studies have emphasised the necessity of re-evaluating the PRISMA-7 scoring system to address concerns pertaining to sex bias, regional variability, and its correlation with validated clinical measures of frailty [29,30,31].

This study demonstrates that PRISMA-6, which excludes the sex-specific Item 2 from PRISMA-7, aligns more closely with CFS, particularly in men, where PRISMA-7 may overestimate frailty due to the added score for the male sex. While Item 2 increases sensitivity in men—who are generally less prone to frailty but may experience more rapid decline and earlier mortality—it may disadvantage women when frailty screening is used to guide preventive resource allocation. Women, despite living longer, often experience a greater loss of healthy life years. The observed sex bias in PRISMA-7 is introduced solely by Item 2 (“Are you male?”) and demonstrates that removing this item—as in PRISMA-6—improves alignment with the CFS in men while leaving results in women unchanged. In addition to scoring algorithms, physiological sex differences, such as smaller absolute muscle mass, respiratory capacity, or body size in females, may influence the interpretation of frailty items, particularly those based on physical capacity or self-report, potentially contributing to differential item functioning between men and women [32]. In this context, PRISMA-6 may offer a more equitable approach for identifying pre-frail individuals in both sexes. However, given these trade-offs, a high-quality validation study of PRISMA-6 is essential before it can be recommended for routine use.

### Strengths and Limitations

This study had several strengths, including its substantial sample size. It was conducted in general practice settings, providing real-world insights into the applicability of frailty screening tools, such as PRISMA-7 and PRISMA-6, in routine clinical care. The inclusion of the CFS as a reference tool confers a significant advantage over previous investigations [11], which relied solely on PRISMA data. However, the restriction to general practice populations may limit the generalisability of these findings to broader community-dwelling populations. Furthermore, the cross-sectional design precludes conclusions regarding the longitudinal predictive value of the tools for adverse outcomes, such as hospitalisation or mortality. While the CFS provides a clinician-judged frailty assessment, it is not the gold standard for frailty diagnosis, which may affect the interpretation of the alignment between the tools. Additionally, PRISMA-7 and PRISMA-6 rely on self-reported data, which may be subject to recall or reporting bias, particularly in older adults with potential cognitive impairments, as cognitive status was not systematically assessed. Finally, the findings are specific to the South Tyrol healthcare setting and may not be directly applicable to other regions with different healthcare systems or demographics.

Future research should validate PRISMA-6 against other frailty assessment tools and explore longitudinal outcomes and the broader applicability of these findings across diverse populations and healthcare settings to enhance the understanding of the long-term implications of frailty screening in primary care settings. This study contributes to the ongoing effort to optimise frailty screening tools, align with national healthcare reforms in Italy [7], and support evidence-based practices in the care of older adults.

## 5. Conclusions

This study elucidates the potential of the PRISMA-6 as an equitable and reliable frailty screening instrument in primary care settings. By eliminating PRISMA-7’s Item 2, which systematically overestimated frailty in male subjects, PRISMA-6 achieved enhanced alignment with the CFS and mitigated sex bias in frailty classification. The findings demonstrate that PRISMA-6 more accurately reflects functional and clinical deficits, thus providing a more balanced frailty classification across the sexes.

However, as this constitutes only the second report on PRISMA-6, the proposition to use it as a screening tool, particularly for male patients, remains conditional and necessitates prospective validation. Longitudinal studies are required to evaluate its predictive accuracy for adverse health outcomes such as hospitalisation, functional decline, and mortality. Furthermore, its performance should be assessed across diverse settings and compared with instruments designed to identify pre-frailty, ensuring its robustness in identifying individuals across the frailty spectrum.

While PRISMA-6 appears to reduce sex-related classification bias and aligns more closely with clinical judgment, further validation is needed before it can be definitively recommended for routine frailty screening.

## Figures and Tables

**Figure 1 diagnostics-15-00915-f001:**
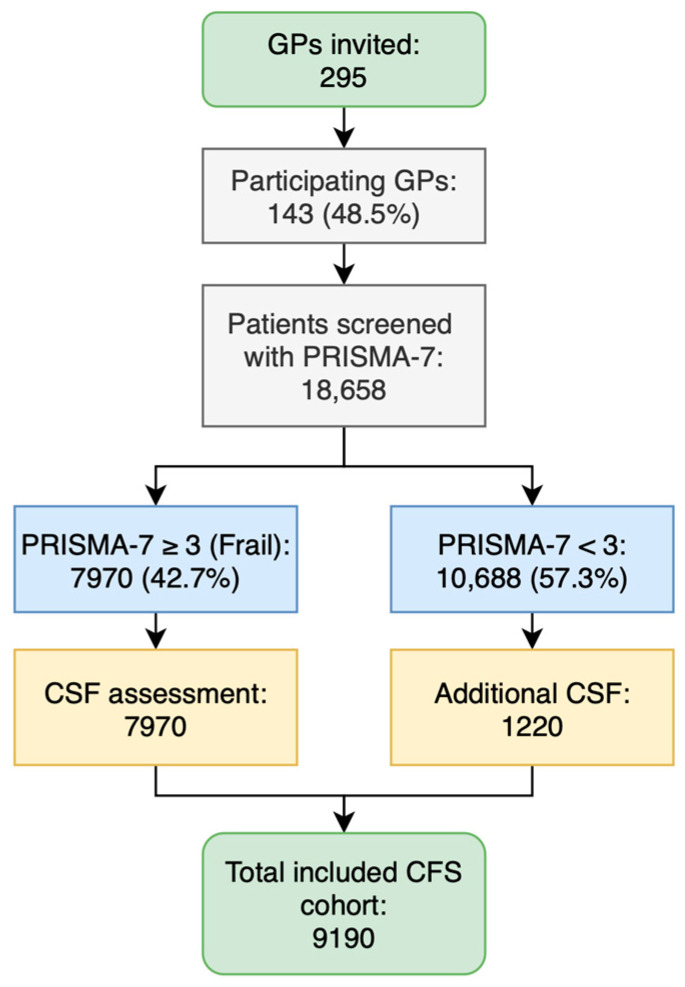
A flowchart of GP participation and patient inclusion in the frailty screening and assessment process.

**Figure 2 diagnostics-15-00915-f002:**
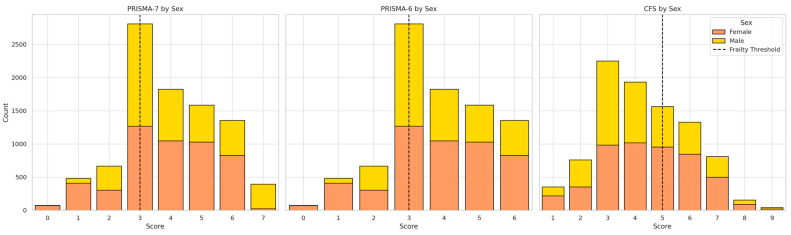
The distribution of frailty scores by sex across three assessment tools. Stacked bar plots showing the distribution of scores for PRISMA-7, PRISMA-6, and the Clinical Frailty Scale (CFS) among 9190 general practice patients aged ≥75 years, stratified by sex. Frailty thresholds: PRISMA-7 and PRISMA-6: score ≥ 3; CFS: score ≥ 5.

**Figure 3 diagnostics-15-00915-f003:**
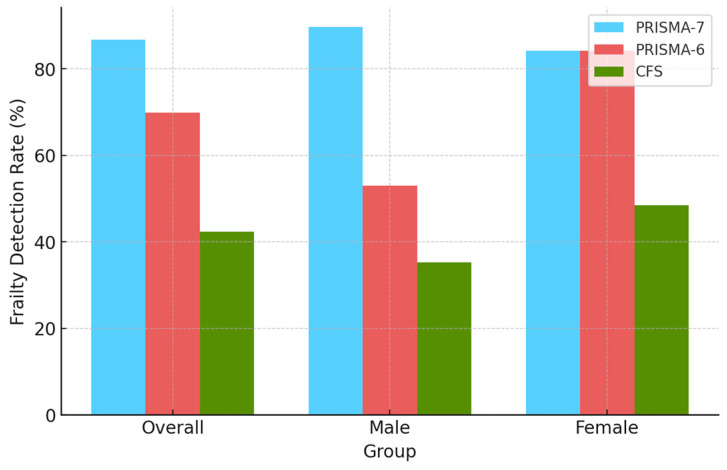
Comparison of frailty detection rates by tools. CFS, Clinical Frailty Scale.

**Table 1 diagnostics-15-00915-t001:** Frailty detection rate according to age, sex, and frailty tool.

Age Group	Sex (*n*)	Tool	Frailty Detection Rate		
			*n*	Percent ^1^ [Lower; Upper CI]	OR ^1^ [Lower; Upper CI]
Overall ^2^	Total (9190) ^3^	PRISMA-7	7970	86.7 [86.03; 87.42]	6.53 [5.75; 7.45]
		PRISMA-6	6425	69.9 [68.98; 70.85]	2.32 [2.14; 2.53]
		CFS	3894	42.4 [41.36; 43.38]	0.74 [0.68; 0.81]
	Male (4212)	PRISMA-7	3777	89.7 [88.75; 90.59]	8.68 [7.12; 10.58]
		PRISMA-6	2232	53.0 [51.48; 54.50]	1.13 [0.99; 1.29]
		CFS	1485	35.3 [33.81; 36.70]	0.54 [0.48; 0.61]
	Female (4978) ^4^	PRISMA-7	4193	84.2 [83.22; 85.24]	5.34 [4.68; 6.10]
		PRISMA-6	4193	84.2 [83.22; 85.24]	5.34 [4.68; 6.10]
		CFS	2409	48.4 [47.00; 49.78]	0.94 [0.84, 1.05]
75–84	Total (4811)	PRISMA-7	3725	77.4 [76.25; 78.61]	3.43 [3.02, 3.90]
		PRISMA-6	2742	57.0 [55.60; 58.39]	1.33 [1.18; 1.50]
		CFS	1507	31.3 [30.01; 32.63]	0.46 [0.41; 0.52]
	Male (2354)	PRISMA-7	1946	82.7 [81.14; 84.20]	4.77 [3.96; 5.73]
		PRISMA-6	963	40.9 [38.92; 42.90]	0.69 [0.58; 0.83]
		CFS	642	27.3 [25.47; 29.07]	0.37 [0.31; 0.44]
	Female (2457) ^4^	PRISMA-7	1779	72.4 [70.64; 74.17]	2.62 [2.26; 3.03]
		PRISMA-6	1779	72.4 [70.64; 74.17]	2.62 [2.26; 3.03]
		CFS	865	35.2 [33.32; 37.09]	0.54 [0.46; 0.63]
85+	Total (4379)	PRISMA-7	4245	96.9 [96.43; 97.45]	31.68 [22.15; 45.28]
		PRISMA-6	3683	84.1 [83.02; 85.19]	5.29 [4.54; 6.15]
		CFS	2387	54.5 [53.04; 55.99]	1.20 [1.06; 1.36]
	Male (1858)	PRISMA-7	1831	98.5 [98.00; 99.09]	67.81 [36.85; 124.80]
		PRISMA-6	1269	68.3 [66.18; 70.42]	2.15 [1.73; 2.67]
		CFS	843	45.4 [43.11; 47.64]	0.83 [0.70; 0.99]
	Female (2521) ^4^	PRISMA-7	2414	95.8 [94.97; 96.54]	22.56 [16.15; 31.54]
		PRISMA-6	2414	95.8 [94.97; 96.54]	22.56 [16.15; 31.54]
		CFS	1544	61.2 [59.34; 63.15]	1.58 [1.35; 1.84]

^1^ The detection rate and odds ratios (ORs) are presented with 95% confidence intervals (CI). The OR quantifies the likelihood of being classified as frail relative to not being classified as frail within the same population. ^2^ General practice patients for whom age, sex, PRISMA-7, and CFS scores were available. ^3^ The Clinical Frailty Scale (CFS) was applied to patients classified as frail by the Program of Research to Integrate the Services for the Maintenance of Autonomy-7 (PRISMA-7) score ≥ 3. ^4^ Since all female participants scored 0 on PRISMA-7 Item 2 (“Are you male?”), the PRISMA-6 and PRISMA-7 scores are identical in this subgroup. Consequently, their correlation with the CFS yields the same value, and comparative analyses between PRISMA-6 and PRISMA-7 are only meaningful for the male subgroup.

**Table 2 diagnostics-15-00915-t002:** Correlations between PRISMA scores and Clinical Frailty Scale (CFS).

Sex (*n*)	Variable		PRISMA-7	PRISMA-6	CFS
Overall (9160)	PRISMA-7	Kendall’s Tau-b ^1^	—		
		*p*-value	—		
	PRISMA-6	Kendall’s Tau-b	0.596	—	
		*p*-value	<0.001	—	
	CSF	Kendall’s Tau-b	0.308	0.492	—
		*p*-value	<0.001	<0.001	—
Male (4112)	PRISMA-7	Kendall’s Tau-b	—		
		*p*-value	—		
	PRISMA-6	Kendall’s Tau-b	0.360	—	
		*p*-value	<0.001	—	
	CSF	Kendall’s Tau-b	0.226	0.576	—
		*p*-value	<0.001	<0.001	—
Female (4987)	PRISMA-7	Kendall’s Tau-b	—		
		*p*-value	—		
	PRISMA-6	Kendall’s Tau-b	1.000	—	
		*p*-value	<0.001	—	
	CSF	Kendall’s Tau-b	0.389	0.389	—
		*p*-value	<0.001	<0.001	—

^1^ All tests were one-tailed for positive correlations.

## Data Availability

Data are available from the corresponding author upon reasonable request.

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
