# Peer review of "Sex Bias in Frailty Screening: A Cross-Sectional Analysis of PRISMA-7 and the Clinical Frailty Scale in Primary Care"

_diagnostics, 2025, doi:10.3390/diagnostics15070915_

Round 1

Reviewer 1 Report

Comments and Suggestions for Authors

Thank you for inviting me to review this manuscript. I hope my comments will improve the overall merits of this study as follows.

  1. Since there are various frailty assessment tools, I recommend authors provide a rationale for selecting PRISMA and CFS for frailty screening and evaluation. It would be beneficial to clearly highlight how these tools compare to others in terms of advantages and limitations.
  2. The conclusion in the abstract should be revised to be more concise and clearly state whether the author recommends using PRISMA-6 for frailty screening in future studies.
  3. Please add sample size calculations to the methods. The ethical approve date and number must be added to the statement too (PAGE10).
  4. There are inconsistent numbers scattering in the context. For example, PAGE 2 “a median of 185 patients each” but PAGE 4 mentioned “190”. Please confirm all numbers presented in the whole manuscript.
  5. An important issue that requires explanation is how the authors managed or controlled interrater variability in CPS assessments across multiple settings. Given the potential variation among evaluators, it is essential to clarify the methods used to ensure consistency. Without such details, the reliability of the study’s findings may be compromised.
  6. PAGE 4 “Since the date were not normally distributed……in such data structures” This statement should be moved to statistical analysis.
  7. Recommending adding the flowchart of participant enrollment. The current description in the first paragraph of PAGE 4 is quite confusing.
  8. Please confirm that the last sentence on PAGE 5 is correct. I think PRISMA-7 showed “inconsistently” higher frailty prevalence…
  9. To ensure the validity of future research findings, the same frailty screening and assessment tools should be used for all participants. The current discussion and conclusion sections suggest that PRISMA-6 is specifically recommended for male participants. However, I am unsure if this interpretation is correct. I suggest revising the discussion for clarity and refining the conclusion to be more concise.

Reviewer 2 Report

Comments and Suggestions for Authors

Reviewer Comments

Thank you for the opportunity to review this timely and well-conducted study. The manuscript addresses an important issue, which is the potential sex bias in PRISMA-7, and proposes a practical, evidence-supported alternative in PRISMA-6. However, further clarifications are required to improve the clarity, transparency, and logic of the manuscript.

1. Introduction

  • Page 2 Line 58= Please clarify in detail the rationale for evaluating Item 2 (“Are you male?”) in PRISMA 7.
  • The current text briefly mentions potential bias but does not explain why and how this item may lead to overclassification in men. What is the scoring system, and how does this inflate the chance of men being diagnosed as frail compared to women? A sentence or two explaining how this item was initially included based on historical associations between male sex and adverse outcomes and why this may no longer be appropriate would strengthen the rationale.

2. Methods

  • While PRISMA-7 and CFS are named, their content and scoring mechanisms are not detailed. Consider briefly listing or summarizing PRISMA-7 items (or adding an appendix) and describing the CFS scale (1–9, with ≥5 indicating frailty).
  • Explain the inclusion of PRISMA-7–negative patients in CFS assessment which was explained in the results (Page 5, Line 154 ) but missing in Methods
    The manuscript reports that 1,220 patients with PRISMA-7 <3 were also assessed with CFS, yet this is not mentioned in the Methods section. Please add a sentence clarifying how these patients were selected.
  • Page 3, Line 100 :The information in this line repeats content already conveyed in the previous sentences. Consider streamlining or rephrasing to improve flow and avoid redundancy.
  • Clarify odds ratio calculation and its purpose: The author explains how ORs were calculated, but it’s unclear what the reference group is in each case. Also, in a high-prevalence context (e.g., frailty rates >80%), ORs can be misleading. Consider briefly justifying their use or supplementing with absolute prevalence rates. I think that prevalence is enough for comparing the detection rate. However, if OR is used, please be clear about how to interpret it and its relation with the prevalence rate

3. Results and Figures

  • Clarify or simplify Figure 1 (score distributions), as the value of the age breakdown is unclear since age-specific differences are not discussed. If not directly analyzed, consider simplifying the figure. Also, clarify the x-axis score ranges for each tool and mark frailty thresholds visually.
  • In Table 2, since all women score 0 on PRISMA-7 Item 2 thus, PRISMA-6 and PRISMA-7 are identical in this group; therefore, comparing its association with CFS appears to be redundant. Consider noting this explicitly in the text or simplifying the table.

4. Interpretation and Discussion

  • The discussion could more directly emphasize that the sex bias is introduced solely by Item 2, and that PRISMA-6 improves alignment with CFS in men without changing results in women.

Overall Recommendation

This is a valuable, clearly structured study with strong implications for equitable screening in older adults. With a few clarifications and adjustments, particularly in the  Finally, I commend the authors for addressing a subtle but important issue in frailty screening and for doing so with thoughtful analysis.

Reviewer 3 Report

Comments and Suggestions for Authors

Round 2

Reviewer 1 Report

Comments and Suggestions for Authors

Authors had carefully edited the manuscripts according to the comments.

Reviewer 3 Report

Comments and Suggestions for Authors

The manuscript is well revised.